# Lameness in Early Lactation Is Associated with Lower Productive and Reproductive Performance in a Herd of Supplemented Grazing Dairy Cows

**DOI:** 10.3390/ani11082294

**Published:** 2021-08-03

**Authors:** Joaquín Chiozza Logroño, Ramiro Rearte, Santiago Gerardo Corva, Germán Ariel Domínguez, Rodolfo Luzbel de la Sota, Laura Vanina Madoz, Mauricio Javier Giuliodori

**Affiliations:** 1Instituto de Investigaciones en Reproducción Animal (INIRA), Facultad de Ciencias Veterinarias, Universidad Nacional de La Plata (FCV-UNLP), La Plata B1900AVW, Argentina; joaquinchiozzavet@gmail.com (J.C.L.); ramirorearte1@gmail.com (R.R.); sgcorva@hotmail.com (S.G.C.); dairydoc82@gmail.com (R.L.d.l.S.); vaninamadoz@gmail.com (L.V.M.); 2Consejo Nacional de Investigaciones Científicas y Tecnológicas (CONICET), Ciudad Autónoma de Buenos Aires C1033AAJ, Argentina; 3Cátedra de Epidemiología, Facultad de Ciencias Veterinarias, Universidad Nacional de La Plata (FCV-UNLP), La Plata B1900AVW, Argentina; 4Actividad Privada, Venado Tuerto, Santa Fe S2600GOZ, Argentina; germandominguez@powervt.com.ar; 5Cátedra de Fisiología, Facultad de Ciencias Veterinarias, Universidad Nacional de La Plata (FCV-UNLP), La Plata B1900AVW, Argentina

**Keywords:** clinical lameness, milk yield, reproductive performance, grazing dairy cow

## Abstract

**Simple Summary:**

It has been reported that the detrimental impact of clinical diseases, such as mastitis, on lactation and reproduction is highest when the first clinical case occurs in early lactation. Therefore, we run an observational study on 7156 lactations from highly supplemented grazing dairy cows to evaluate the association of the timing of lameness case occurrence in lactation with productive and reproductive performances in dairy cows. We found that cows getting lame before the first service produced less milk than cows getting lame later in lactation (i.e., after the first service), and that both groups of lame cows produced less milk than healthy ones. We also found that cows becoming lame after the first service had an 87 d longer calving to pregnancy interval than healthy herd mate cows and that cows turning lame before the first service had an 38 d longer calving to pregnancy interval than healthy herd mates. In conclusion, the timing of lameness case occurrence in lactation is associated with its impact on productive and reproductive performances in dairy cows.

**Abstract:**

The main aim of this study was to assess the associations between the timing of lameness clinical case occurrence in lactation with productive and reproductive performances in grazing Holstein cows. A cohort study was carried out on a dataset with records from a commercial dairy herd (Buenos Aires, Argentina) for cows that calved and were dried off from January 2010 through June 2017. The first recorded event of lameness per lactation was considered for the study. Criteria for lactation inclusion included not having uterine diseases, mastitis, or anovulatory cysts during the studied risk period (i.e., up to 200 DIM). Therefore, a total of 7156 out of 20,086 lactations were included in the statistical analysis. The association between lameness case occurrence in lactation (cows not lame (LG0) vs. lame cows between parturition and first service (LG1) vs. lame cows between first service and first pregnancy (LG2)) with productive (i.e., accumulated milk yield to 150 DIM (MILK150) and 300 DIM (MILK305)) and reproductive performances (hazard of insemination and pregnancy) was analyzed with linear regression models and proportional hazard regression models, respectively. Lame cows produced 161 and 183 kg less MILK150 and MILK305 than non-lame herd mates, respectively. Moreover, LG1 cows produced 216 kg less MILK150 and 200 kg less MILK305 than LG0 cows, and LG2 cows also produced 58 kg less MILK150 and 158 kg less MILK305 than LG0 cows. The LG1 cows had a lower hazard of service than LG0 cows (HR = 0.43, 95%CI = 0.39–0.47). Furthermore, LG1 cows had a lower hazard of pregnancy than LG0 cows (HR = 0.52, 95%CI = 0.46–0.59) and took longer to get pregnant than LG0 cows (median [95%CI], 139 [132–144] vs. 101 [99–103]). Moreover, LG2 cows had a much lower hazard of pregnancy than LG0 cows (HR = 0.08, 95%CI = 0.05–0.12) and much longer calving to first pregnancy interval than LG0 cows (188 [183–196] vs. 101 [99–103]). In conclusion, cows that become lame in early lactation produce less milk and have lower hazards of insemination and pregnancy than herd mates that are healthy or become lame later in lactation. In addition, cows that become lame immediately after the voluntarily waiting period have the poorest reproductive performance (i.e., they have the lowest hazard of pregnancy and the longest calving to pregnancy interval).

## 1. Introduction

The economic success of dairy producers depends heavily on reproductive and productive performances. Many reports showed that production diseases, such as lameness, mastitis, and infertility, negatively impact milk yield and reproductive performance [1,2]. Regarding lameness, it is known that its worldwide prevalence can range from values as low as <10% up to as high as >50% [3,4,5,6]. Most of the reports about the association of clinical lameness with milk yield and reproductive performance are from big-frame and high-producer cows (i.e., American Holstein type) allocated in confined systems with total mixed rations (TMRs) [7,8,9,10]. There are some reports coming from non-seasonal calving pasture-based U.K. systems with mid-producing Holstein–Friesians cows (7300 to 9400 kg of MILK305) [11]; from winter-housed, spring-calving pasture-based U.K. systems with low producing (6400 kg of MILK305) Holstein, Jersey, and Friesian cows [12]; and from seasonal calving pasture-based New Zealand systems with low producing mixed breed cows (Holstein–Friesian, Jersey, and Holstein–Friesian × Jersey crossbred) [13] or small-frame Friesian cows [14]. However, there is a lack of data from mixed systems like the Argentinean, where big-frame cows are reared under supplemented non-seasonal grazing conditions and kept outdoors year-round.

Therefore, our working hypothesis states that the negative effect of clinical lameness on milk yield and reproductive performance depends on the timing of disease occurrence in lactation. Thus, to test that hypothesis, our main objective was to assess the association between the timing of clinical lameness case occurrence in lactation with productive and reproductive performances in grazing dairy cows supplemented with partial mixed rations (PMRs).

## 2. Materials and Methods

### 2.1. Study Farm and Herd Management

This study was performed on a commercial dairy farm selected because of its long-standing relationship with our research group. The farm, located in Carlos Casares (35°37′ S, 61°22′ W), Buenos Aires province, Argentina, had approximately 2600 milking cows. Rolling herd average milk production was approximately 11,818 kg. Prepartum transition cows within 4 weeks of the expected calving date were kept on dry lots, fed a low dietary cation-anion difference (DCAD) diet, and monitored for signs of calving by farm employees trained to assist with parturition. After calving, cows were sent for 3 d to the fresh herd and kept on a dry lot. At 4 d postpartum, healthy cows were moved to a lactating herd. Lactating cows were at pasture in a rotational system (different paddocks in the morning and afternoon). Feed was composed of mixed pastures (alfalfa, tall fescue) and winter annual grasses (ryegrass), and concentrates (40% soybean pellets and 60% cornmeal) were offered twice daily during milking and supplemented with partial mixed ration (PMR) diets (corn silage, soybean pellets, and cornmeal) formulated to meet or exceed the National Research Council (NRC, 2001) nutrient requirements for lactating Holstein cows weighing 650 kg and producing 45 kg of 3.5%. Cows were milked twice a day (04:00 and 16:00), and milk yield was recorded during the official monthly milk test.

### 2.2. Reproductive Management

Breeding occurred year-round except for the hot summer months (January and February). The farm was visited by a veterinarian (GAD) every 14 days. Cows were observed for signs of estrus twice daily using tail chalking [15] after a voluntary waiting period of ~50 days. Cows were inseminated when detected in estrus. Cows not found in estrus by 70 DIM were subjected to a fixed-timed AI program. Pregnancy diagnosis was performed 30–45 days after AI by ultrasonographic examination with a 7.5 MHz linear transrectal transducer. Pregnancy was confirmed by ultrasonographic visualization of a live embryo. Non-pregnant cows were re-synchronized and inseminated at a fixed time.

### 2.3. Lame Cow Management

Lameness diagnosis was performed by the farm personnel, trained by the farm veterinarian (GAD), immediately after milking every two weeks. A cow was considered lame when having a locomotion score of ≥4 (five-point scale, [16]). Lame cows were hoof trimmed by trained farm personnel and received a systemic antibiotic drug when lesions compatible with foot rot were observed and kept in a pen near the milking parlor until recovery (locomotion score ≤3). Finally, foot bathing was carried out at three consecutive milkings per week for lactating cows and weekly for dry cows using a 5% formalin solution.

### 2.4. Dataset Management and Statistical Analysis

A cohort study was carried out on a dataset with records from this commercial dairy herd for cows that calved and were dried off from January 2010 through June 2017. Data were extracted from commercial software (Protambo Master 3.5; DIRSA S.A., Gonnet, Argentina). The first recorded event of lameness per lactation was considered for the study. Criteria for lactation inclusion were not having uterine diseases, mastitis, or anovulatory cysts during the risk period studied (i.e., up to 200 DIM). Additionally, lactations should have a record of body condition score (BCS) measurement around calving. Therefore, a total of 7156 out of 20,086 lactations were included in the statistical analysis.

### 2.5. Lameness and Milk Yield

The associations between lameness and productive performance (accumulated milk yield to 150 (MILK150) and 300 DIM (MILK305)) were analyzed with mixed linear regression models using the Proc GLIMMIX of SAS with normal distribution and identity link function. Univariable linear models were run first and those predictors having *p* ≤ 0.25 were offered to multivariable linear models where they remain if *p* < 0.1. A backward elimination process (one at a time) was used to remove predictors having *p* > 0.1. Postpartum BCS (a potential confounder) was forced to remain in the models. The models included the random effect of the cow’s lactation and the fixed effects of lameness (no vs. yes). The models were controlled for year of calving (2010 through 2018), season of calving (summer (21 December to 20 March), fall (21 March to 20 June 20), winter (21 June to 20 September 20), and spring (21 September to 20 December)), parity (1st vs. 2nd vs. ≥3rd), and postpartum BCS (≤2.50 vs. 2.75–3.25 vs. ≥3.50). First-order interactions were also tested. In addition, the association between the timing of lameness case occurrence in lactation with productive performance (MILK150 and MILK305) was also analyzed with mixed linear regression. The model included the random effect of the cow and the fixed effect of lameness occurrence categorized as follows: cows not diagnosed as lame during the entire lactation (healthy cows, LG0), cows diagnosed as lame between parturition and first service (LG1), and cows diagnosed as lame between first service and first pregnancy (LG2). An orthogonal contrast was used to test LG0 versus LG1 and LG2, and LG1 versus LG2.

### 2.6. Lameness and Reproductive Performance

The associations between the timing of the lameness case occurrence in lactation (LG0 vs. LG1 vs. LG2) with first service and pregnancy hazard were assessed with Proc PHREG of SAS. Proportional hazard regression models also included year as blocking factor and season of calving and parity as fixed effects. Modeling was performed as described above. Proportionality of the hazards was checked by the Kaplan–Meier survival analysis assuming a predictor satisfies the proportional hazard assumption when the graph has parallel curves. Time intervals (median and 95%CI) were obtained with Kaplan–Meier survival analysis using PROC LIFETEST of SAS.

Statistical significance was set at *p* < 0.05. The cow was considered the experimental unit.

## 3. Results

### 3.1. Lameness and Milk Yield

Lameness was negatively associated with milk yield, given that lame cows produced 161 and 183 kg less of MILK150 and MILK305 than non-lame herd mates, respectively (Table 1). The timing of lameness case occurrence in lactation was also negatively associated with milk yield in grazing dairy cows, given that LG1 cows produced 216 kg less MILK150 and 200 kg less MILK305 than LG0 cows, and LG2 cows also produced 58 kg less MILK150 and 158 kg less MILK305 than LG0 cows (Table 2).

### 3.2. Lameness and Reproductive Performance

Lame cows between parturition and first service (LG1) had a lower hazard of service than non-lame herd mates (HR = 0.43, 95%CI = 0.39–0.48, *p* < 0.001, Figure 1).

The timing of lameness case occurrence in lactation was associated with the hazard of pregnancy, given that LG1 cows had a lower hazard of pregnancy (HR = 0.52, 95%CI = 0.46–0.59, *p* < 0.001) than LG0 herd mates and had a longer median [95%CI] calving to pregnancy interval (139 [132–144]) than LG0 cows (101 [99–103], Figure 2). Moreover, LG2 cows had a much lower hazard of pregnancy than LG0 herd mates (HR = 0.08, 95%CI = 0.05–0.12) and much longer calving to first pregnancy interval (188 [183–196]) than LG0 cows (101 [99–103], Figure 2). Finally, LG2 cows had a lower hazard of pregnancy than LG1 cows (HR = 0.53, [0.46–0.62], *p* < 0.001, Figure 2) and a longer calving to pregnancy interval (188 [183–196]) than LG1 cows (139 [132–144]).

## 4. Discussion

The finding that lameness is negatively associated with milk yield in grazing dairy cows, given that lame cows produced less MILK150 (i.e., from 58 up to 216 kg) and less MILK305 (i.e., from 158 up to 200 kg), agrees with a previous report showing that lame cows produced 152 to 204 kg less 305 day mature equivalent milk than healthy cows [17]. Other studies have reported higher losses than us. For example, one of them found that severely lame cows in the first month of lactation produce 350 kg (95% CI: 81 to 620) less 305 d milk than non-lame herd mates [11]. Others showed that lame cows at 150 DIM have a reduction of 357 kg (95% CI: 163 to 552) in 305 day lactation [8], and that lame cows diagnosed by farmers with sole ulcer and white line disease produce (mean [95% CI]) 574 [307–841] and 369 [137–600] kg less milk than herd mates, respectively [7]. Even though comparison among studies is difficult owing to differences in lameness definition, cow’s breed, production system, and data analysis, most reported milk losses range between 270 and 574 kg [18]. Therefore, we find a negative association between lameness and milk yield in line with previous reports [7,10,11,19].

An interesting finding regarding the association of the timing of case occurrence in lactation with milk yield is that cases occurring before 1st service compared with those occurring after 1st service have a more significant deleterious effect on MILK150, but not on MILK305. It has been proposed that MILK305 is less sensitive than repetitive daily milk yield owing to lack of temporal association between a lameness event and milk yield [8,20]. In addition, severely lame cows (chronic or recurrent) in the first trimester of lactation are more likely to have lower MILK305 than never lame herd mates [11]. However, non-persistent lame cows in early lactation have a MILK305 that decreases toward that for average never lame cows, so that they partially lost their full potential for a high milk yield [11]. These authors concluded that the largest reductions in MILK305 are associated with cases of persistent severe lameness in early lactation [11]. Therefore, that could explain why we found an association with MILK150, but not with MILK305. Therefore, our results do not support our working hypothesis stating that the negative effect of clinical lameness on milk yield depends on the timing of disease occurrence in lactation. Another finding is that the negative impact of lameness on reproductive performance depends on the timing of case occurrence in lactation with cases diagnosed before the first service (i.e., LG1) and especially those detected between the first service and first pregnancy (i.e., LG2), having 38 and 87 d longer calving to conception intervals than non-lame herd mates, respectively. A previous study showed that cows having a locomotion score of ≥4 during the first 70 DIM have a 31 d longer (158 d vs. 127 d) calving to conception interval and a lower hazard of pregnancy (HR: 0.76, 95%CI: 0.58–0.94) than cows having a locomotion score of <4 [9]. Another study reported that mildly and severely lame cows have a lower hazard of pregnancy (HR: 0.82 and 0.74, respectively) compared with non-lame cows. Moreover, mildly lame cows have a 50 day longer calving to conception interval than non-lame cows (180 vs. 130 days), and severely lame cows have a 66 day longer interval than non-lame cows (200 vs. 134 days [10]). Other researchers found that lame cows have a hazard of conception of 0.78 (95% CI: 0.68–0.86) and take 12 days longer (40 vs. 28 days) to get pregnant compared with non-lame cows in seasonally breeding cattle in pasture-based systems [13]. This delay in conception (12 days) means that lame cows have shorter lactations than non-lame cows in seasonally calving systems where lactation length is determined by feed availability [13]. Finally, another study assessed the fertility of cows detected to be lame before and/or after 1st service, which determined four categories: non-lame before and after 1st serve (NN); lame cows before, but not after 1st service (YN); non-lame cows before 1st service, getting lame after 1st service (NY); and lame cows before and after 1st service (YY). The hazards for pregnancy decreased by a factor 0.88 (95%CI: 0.62–1.29), 0.65 (95%CI: 0.52–0.81), and 0.62 (95%CI: 0.42–0.93) for YN cows, NY cows, and YY cows, respectively, compared with NN cows [14]. Therefore, despite that comparison among studies is difficult, as mentioned above, our results agree with previous reports showing that lame cows have lower hazards of pregnancy (i.e., HR = 0.74 to 0.78 [9,11,21]) and longer calving to conception intervals (i.e., 31 to 50 d) than non-lame herd mates [9,21]. Again, our main finding is that the negative impact of lameness on reproductive performance depends on the timing of case occurrence in lactation. It depends on the temporal association of lameness with breeding, which supports our working hypothesis.

A possible explanation for the lower milk yield and poorer reproductive performance observed in lame cows compared with non-lame herd mates could be that lame cows experience considerable discomfort and pain [22] that lasts for a long time (i.e., before diagnosis up to even after treatment) [8,22]. Therefore, this inflammatory response, elicited by lameness, triggers the release of proinflammatory mediators leading to altered behavioral patterns such as lower eating time [23,24], reduced number of daily meals [23], lower rumination bouts [25], lower intensity of estrus expression, and shorter standing time to be mounted [25]. These behavioral changes lead to reduced DMI and milk yield [23] and deeper NEB, impairing ovarian activity [26]; delaying the return to cyclicity after parturition [27]; reducing oocyte competence [28]; disrupting the uterine environment [29]; and, finally, impairing embryo development and survival [30,31,32]. In addition to behavioral changes, the release of proinflammatory mediators induces insulin and IGF-1 resistance, altering nutrient partitioning by driving energy to support immune cells and away from productive and reproductive performances [33,34].

The main strength of this study is the association between the timing of lameness case occurrence in lactation with productive and reproductive performances in supplemented grazing dairy cows under reproductive management based on almost continuous service. The main limitations are that this cohort study was carried out in only one commercial dairy farm; that the causes of clinical lameness were not registered; that bias could have occurred given that farm personnel tend to underestimate the proportion of affected animals [4]; and, finally, that lameness scores of 3 (i.e., subclinical cases) were not evaluated.

## 5. Conclusions

In conclusion, lame cows produce less milk than healthy herd mates, but the timing of lameness case occurrence in lactation (up to 200 DIM) seems not to be critical to determine the negative association with 305 d milk yield in supplemented grazing dairy cows. However, the timing of lameness case occurrence in lactation is critical to determine the negative association with reproductive performance in grazing dairy cows, given that cows that become lame before the first service have lower hazards of insemination than healthy herd mates, and that cows that become lame after the first service have the most significant negative impact on reproductive performance (i.e., they have the lowest hazard of pregnancy and the longest calving to pregnancy interval). Our results with big-frame dairy cows reared under supplemented non-seasonal grazing conditions and kept outdoors year-round are comparable with other reports from different management systems such as high-producing, big-frame cows confined in American systems; mid-producing cows in non-seasonal calving pasture-based U.K. systems; low-producing cows from winter-housed and spring-calving pasture-based U.K. systems; and low-producing cows from seasonal calving pasture-based New Zealand systems.

## Figures and Tables

**Figure 1 animals-11-02294-f001:**
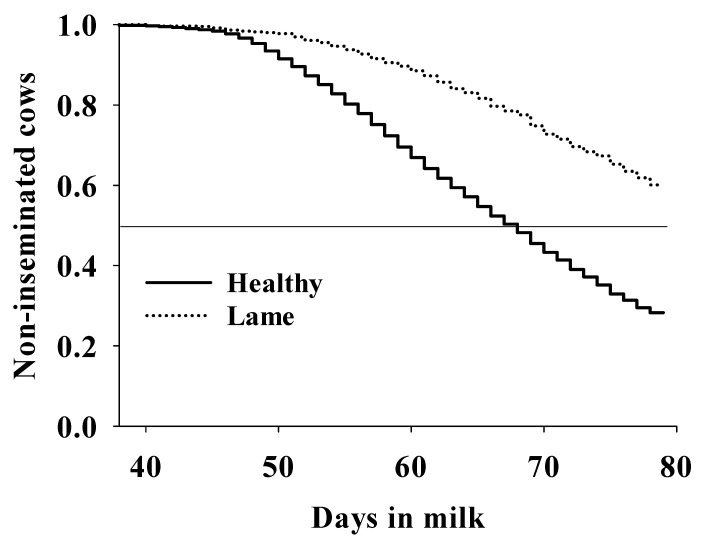
Association between lameness and calving to first service interval in grazing dairy cows (n: 7156) from a commercial dairy herd evaluated for 7.5 years (January 2010–June 2017). Healthy cows had a median (95%CI) calving to the first service interval of 69 (68–69) days; median survival days were not available for lame cows because they still had more than 50% of cows not inseminated by the end of data collection. Lame cows had a lower hazard of the first service than healthy herd mates (HR = 0.43, 95%CI = 0.39–0.48, *p* < 0.001). Lame: cows diagnosed as lame between calving and first service. Cows were diagnosed as lame when having a locomotion score of ≥4 [16]; healthy: cows not diagnosed as lame during the entire lactation.

**Figure 2 animals-11-02294-f002:**
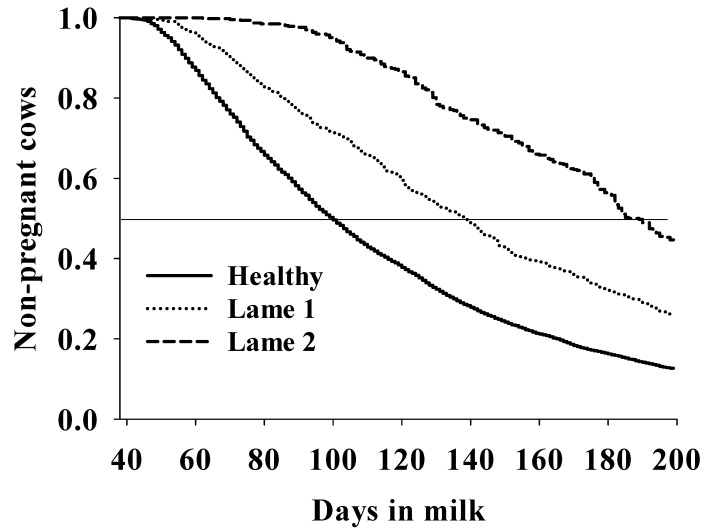
Association between the timing of lameness case occurrence in lactation with calving to pregnancy interval in grazing dairy cows (n: 7156) from a commercial dairy herd evaluated for 7.5 years (January 2010–June 2017). Healthy cows had a median (95%CI) calving to first pregnancy interval of 101 (99–103) days; lame 1 cows had an interval of 139 (132–144) days, and lame 2 cows had calving to first pregnancy interval of 188 (183–196) days. Lame 1 cows had a lower hazard of first pregnancy than healthy herd mates (HR = 0.52, 95%CI = 0.46–0.59, *p* < 0.001), and lame 2 cows also had a lower hazard of first pregnancy than healthy herd mates (HR = 0.08, 95%CI = 0.05–0.12, *p* < 0.001). Finally, LG2 cows had a lower hazard of pregnancy than LG1 cows (HR = 0.53, [0.46–0.62], *p* < 0.001). Healthy: cows not diagnosed as lame during the entire lactation; lame 1: cows diagnosed as lame between calving and first service; lame 2: cows diagnosed as lame between first service and first pregnancy; cows were diagnosed as lame when having a locomotion score of ≥4 [16].

**Table 1 animals-11-02294-t001:** Linear regression models assessed the association between lameness with milk yield in grazing dairy cows (*n* = 6685 lactations for MILK150 and 4298 lactations for MILK305) from a commercial dairy herd evaluated for 7.5 years (January 2010–June 2017).

		MILK150 ^1^				MILK305 ^2^			
		*n*	LSM ^3^	95% CI ^4^	*p*	*n*	LSM ^3^	95% CI ^4^	*p*
Lameness ^5^					<0.001				<0.001
	No	4789	5037	5006–5068		2927	9766	9703–9829	
	Yes	1896	4876	4827–4924		1371	9583	9491–9675	

^1^ MILK150: accumulated milk yield to 150 DIM; ^2^ MILK305: accumulated milk yield to 305 DIM; ^3^ LSM: least squared means were estimated with Proc Glimmix of SAS using normal distribution and identity link function; ^4^ 95% CI: 95% confidence intervals. The models were also controlled by year (2010 through 2018), season (summer (21 December to 20 March 20), fall (21 March to 20 June 20), winter (21 June to 20 September 20), and spring (21 September to 20 December 20)), and parity (1st vs. 2nd vs. ≥3rd). ^5^ Lameness: a case of lameness was defined as cows having a locomotion score of ≥4 [16].

**Table 2 animals-11-02294-t002:** Linear regression models assessed the association between the timing of lameness case occurrence in lactation milk yield in grazing dairy cows (*n* = 6685 lactations for MILK150 and 4298 lactations for MILK305) from a commercial dairy herd evaluated for 7.5 years (January 2010–June 2017).

		*n*	LSM ^1^	95% CI ^2^	*p*	Contrast 1 ^3^	Contrast 2 ^4^
			**MILK150 ^6^**				
Lame group ^5^					<0.001		
	LG0	4789	5037	5006–5037		<0.001	
	LG1	989	4821	4763–4821			<0.001
	LG2	469	4979	4903–5055			
			**MILK305 ^7^**				
Lame group ^5^					<0.001		
	LG0	2927	9766	9703–9829		<0.001	
	LG1	732	9566	9452–9680			0.855
	LG2	416	9608	9472–9744			

^1^ LSM: least squared means were estimated with Proc Glimmix of SAS using normal distribution and identity link function; ^2^ 95% CI: 95% confidence intervals; ^3, 4^ orthogonal contrasts (1: LG0 vs. LG1 and LG2, and 2: LG1 vs. LG2); ^5^ lame group: cows not diagnosed as lame during the entire lactation (healthy cows, L0), cows diagnosed as lame between calving and first service (L1), and cows diagnosed as lame between first service and first pregnancy (L2).^6^ MILK150: accumulated milk yield to 150 DIM; ^7^ MILK305: accumulated milk yield to 305 DIM. The models were also controlled by year (2010 through 2018), season (summer (21 December to 20 March), fall (21 March to 20 June), winter (21 June to 20 September), and spring (21 September to 20 December)), and parity (1st vs. 2nd vs. ≥3rd).

## Data Availability

The data presented in this study are openly available in Mendeley Data at doi: 10.17632/thb3rmxxr6.1.

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
