# Peer review of "Lameness in Early Lactation Is Associated with Lower Productive and Reproductive Performance in a Herd of Supplemented Grazing Dairy Cows"

_animals, 2021, doi:10.3390/ani11082294_

Round 1

Reviewer 1 Report

Good manuscript. We need more information on lameness such as kind of lameness and detection by veterinarians and farmers.

Line

comment

21

There is a study, but only in German with English abstract: Hässig M., Degen Aguayo Aparicio C., Nuss K. Auswirkungen von Lahmheiten zum Zeitpunkt des Trockenstellens auf die Milch- und Fruchtbarkeitsleistungen der folgenden Laktation. Schweiz. Arch. Tierheilk. 160, 115-122, 2018 https://doi.org/10.17236/sat00148

84

Explain abbreviations such as NRC, PMR, DCAD the first time used

171

skip ; and add an empty line

173

add the information of lameness odds per year

205

If you show Fig. 1and 2 with a survival analysis according to Kaplan-Meier, you also have to use appropriate statistics for none-continuous-curves such as Cox proportional hazard analysis. This also has to be mentioned in M&M.

278

There is an important bias which must be addressed. There are several publications on detecting lameness in bovines, which shows that only 60% of all lameness is detected by the farmer. These regression to the mean bias has to be addressed

Reviewer 2 Report

Two major issues:
One you have not read and cited Archer et al (2010)

Your discussion is just a generic repetition of why people think lameness causes milk loss - it needs to be a discussion of where your data fits in the research published on milk loss and reproduction after lameness.

Further comments on attached document

Round 2

Reviewer 2 Report

Thank you for revisions. I still have major concerns which are outlined on the attached file

Round 3

Reviewer 2 Report

Thank you very much for your revisions. I am much happier with the paper. I have a few minor concerns/comments that I have put on attached PDF
